# Integrating Structured and Unstructured EHR Data for Predicting Mortality by Machine Learning and Latent Dirichlet Allocation Method

**DOI:** 10.3390/ijerph20054340

**Published:** 2023-02-28

**Authors:** Chih-Chou Chiu, Chung-Min Wu, Te-Nien Chien, Ling-Jing Kao, Chengcheng Li, Chuan-Mei Chu

**Affiliations:** 1Department of Business Management, National Taipei University of Technology, Taipei 106, Taiwan; 2College of Management, National Taipei University of Technology, Taipei 106, Taiwan

**Keywords:** structured vs. unstructured data, machine learning, intensive care units, electronic health records, predictive modeling

## Abstract

An ICU is a critical care unit that provides advanced medical support and continuous monitoring for patients with severe illnesses or injuries. Predicting the mortality rate of ICU patients can not only improve patient outcomes, but also optimize resource allocation. Many studies have attempted to create scoring systems and models that predict the mortality of ICU patients using large amounts of structured clinical data. However, unstructured clinical data recorded during patient admission, such as notes made by physicians, is often overlooked. This study used the MIMIC-III database to predict mortality in ICU patients. In the first part of the study, only eight structured variables were used, including the six basic vital signs, the GCS, and the patient’s age at admission. In the second part, unstructured predictor variables were extracted from the initial diagnosis made by physicians when the patients were admitted to the hospital and analyzed using Latent Dirichlet Allocation techniques. The structured and unstructured data were combined using machine learning methods to create a mortality risk prediction model for ICU patients. The results showed that combining structured and unstructured data improved the accuracy of the prediction of clinical outcomes in ICU patients over time. The model achieved an AUROC of 0.88, indicating accurate prediction of patient vital status. Additionally, the model was able to predict patient clinical outcomes over time, successfully identifying important variables. This study demonstrated that a small number of easily collectible structured variables, combined with unstructured data and analyzed using LDA topic modeling, can significantly improve the predictive performance of a mortality risk prediction model for ICU patients. These results suggest that initial clinical observations and diagnoses of ICU patients contain valuable information that can aid ICU medical and nursing staff in making important clinical decisions.

## 1. Introduction

The World Federation of Societies of Intensive and Critical Care Medicine defines an intensive care unit (ICU) as an organized system of care for critically ill patients that provides intensive and specialized medical and nursing care, enhanced monitoring capabilities, and multiple physiological organ support modality to sustain life during a period of multiple organ dysfunction syndrome (MODS) [1]. The hospital has established an intensive care unit (ICU) for patients with severe or life-threatening conditions. ICU mortality and costs are the highest of all hospital units [2]. It is difficult for medical and nursing staff to deal with rapidly changing patient conditions if there is not enough real-time information for clinicians to make accurate and timely decisions [3]. Different types of judgment errors can have many negative consequences, and incorrect decisions or delayed diagnosis can have a significant impact on patient prognosis, medical resource availability, and healthcare costs [4]. Recently, when the COVID-19 pandemic flooded intensive care units around the world, their significance was highlighted. In times such as these, more active research on how to manage scarce critical care resources is required to provide additional tools to support medical decision-making and effective clinical practice benchmarks [5]. In the United States, more than 5 million patients are admitted to the ICU annually, and 40% of these patients die during their hospital stay, with 22% spending their entire hospital stay in the ICU [6]. Predicting mortality in ICU patients is one of the most important tasks in critical care research, not only to aid health professionals in clinical decision-making, but also as a basis for managing hospital resource utilization. Patients admitted to the ICU require close and constant monitoring to prevent rapid deterioration of their health. Intensive monitoring through ICU equipment generates a large number of medical records, requiring an efficient and accurate data analysis system [7].

The electronic health record (EHR) is a digital version of the paper chart. Numerous researchers have utilized EHR database data in the past to predict patient mortality, admission time, disease diagnosis, disease onset, etc., to prevent and intervene in early disease in patients which is crucial to critical care. As an essential risk assessment tool, the predictive model has been developed and utilized in numerous healthcare fields. The Sequential Organ Failure Assessment (SOFA), a new Simplified Acute Physiology Score (SAPSII), and the Multiple Organ Dysfunction Score (MODS) have also been used widely in clinical practice to predict mortality [8,9,10]. Predictive models facilitate the early identification of patients at risk for a disease or event and provide effective intervention measures for those who are most likely to benefit from the identification of specific risk factors. Much research has been conducted to determine how data analysis and prediction can assist medical and nursing staff in the process of diagnosis and treatment to heighten alertness to the progression of patient condition [11,12,13,14,15]. The results of the statistical data’s predictive power derived from the basic vital signs and simple demographic data such as age save the most resources and are the most useful. Vital signs were chosen as features mainly because most vital signs can be easily measured using non-invasive equipment, and vital signs are the most basic health indicators that are easily understood by all healthcare professionals [16,17,18,19].

Unstructured data comprise 80% of EHR data [20]. It is undeniable that overlooking the deficiencies of qualitative data in the EHR may not only result in the omission of key factors caused by the absence of handwritten diagnostic data, but may also result in the omission of clues in the initial judgement being overlooked or diminished. Although these variables can be used to partially predict the mortality of ICU patients, quantitative variables are utilized in the majority of these studies. After all, the existing statistical predictive modeling is relatively mature with respect to the processing of quantitative data, whereas distinctive challenges exist in the standardization and utilization of qualitative data [21,22,23].

Machine learning is a subfield of artificial intelligence concerned with teaching computers to learn from data and improve with experience. It focuses on the issue of how to design computer programs that can automatically improve output accuracy based on experience [24]. There has recently been an increase in the use of machine learning applications in clinical medicine. These include preclinical data processing, bedside diagnostic assistance, patient stratification, treatment decision-making, and early warning for primary and secondary prevention [25]. Machine learning can improve clinical decision-making in many ways, by providing early warning, facilitating diagnosis, conducting widespread screening, personalizing treatment, and assessing patient response to treatment. Many different fields and clinical applications are gradually adopting machine learning from mature preclinical scenarios [26,27]. 

The development of machine learning includes text mining, natural language processing and Latent Dirichlet Allocation (LDA) which are used to identify and extract information or relationships from unstructured data and have become popular techniques for literary analysis [28,29]. LDA is a Bayesian probability generation model in the field of natural language processing proposed by Blei et al. [30] that has several advantages for literature analysis. LDA is a powerful tool for processing massive amounts of data that can capture text-specific dimensions without relying on assumptions. Furthermore, it incorporates multiple steps of text analysis, such as data sampling with minimal human intervention to yield more realistic and objective topic modeling outcomes [31,32]. 

However, it takes a lot of time and money to process the unstructured data that make up medical big data. This is particularly so for the digital part, typically a vital component, presented in the large number of clinical notes made during treatment and hospital stay. In accordance with the rules of unstructured data processing, numbers are frequently removed to reduce their utility. By incorporating unstructured data as input, we are not using raw physiological data, but rather the perception and judgment of medical and nursing professionals in the form of free text annotations. These allow us to access higher-level concepts that are not present in the physiological data. The text data format is relatively consistent, and this allows circumvention of the LDA digital deletion limitation. This is the most noticeable feature of free text records, which contain information about patients’ admission to and diagnosis in the ICU. Data about observations and first signs of condition and diagnoses are added as soon as possible after admission of the patient to the ICU, with minimal interference from the earlier patient data. Clinicians can also use the topic obtained as a follow-up reference. Our recent study combined 16 structured variables and 10 topic modeling semi-structured variables from the Medical Information Mart for Intensive Care (MIMIC-III) dataset to predict mortality in ICU patients. The results show that semi-structured data contain useful information that can help clinicians make critical clinical decisions [33].

In this study, we utilized the MIMIC-III database to develop a model for predicting mortality in ICU patients. Our approach involved integrating structured data, which are basic and easily collected from ICU patients, with unstructured data derived from the initial clinical diagnosis of the patient’s physician at the time of admission. We used the LDA approach to topic modeling of diagnostic records and applied machine learning techniques to combine both structured and unstructured data to build a robust mortality risk prediction model. This model can provide patients, their families, and healthcare professionals with valuable additional information for making informed medical decisions. Our findings could have significant implications for improving patient outcomes and advancing critical care medicine.

## 2. Materials and Methods

### 2.1. Proposed Framework

Figure 1 depicts the framework of this study. The structured data collected after patients were admitted to the ICU was integrated (six vital sign measurements in the first 24 h, the Glasgow Coma Scale (GCS), and patient age) with unstructured data (initial clinical diagnosis records at ICU admission). The machine learning model was used to predict mortality of the ICU patient. Finally, five different metrics were used to assess predictive performance. The period of mortality in ICU patients is defined as follows:

### 2.2. Data Collection and Preprocessing 

The rapid development of digital health systems has occurred in recent years. However, concerns surrounding personal privacy and security have made it difficult to integrate and apply this information to scientific research. To ensure the convenience and completeness of data collection, this study has focused on obtaining complete patient dynamic information from databases that are easier to obtain than ICU data. The data were obtained from MIMIC-III clinical database in our research. MIMIC-III uses integrated comprehensive clinical data from patients admitted to the Beth Israel Deaconess Medical Center in Boston, Massachusetts [34]. The MIMIC-III database used contained information on 46,520 patients and 58,976 admission-related data items, including patient vital signs, drugs, laboratory measurement values, and observation records. There were 38,597 adult patients, 56% of whom were male, and the median age was 65.8. The median of the length of admission was 6.9 days, the mortality rate during admission was 11.5%, and the median of the length of stay in the ICU was 2.1 days. Furthermore, the following data were generated per patient per stay in the intensive care unit: 6643 patient observation records, 83 patient medical document records, and 559 laboratory test result records. Table 1 shows the database compilation. The National Institutes of Health (NIH) online course was completed, as well as an exam for protecting human research participants and the submission of an access application (Certification Number: 35628530).

To reflect the universality of the analytical results and to ensure that they were comparable with the conclusions of the related literature, this study followed the patient selection principles of previous related studies and specific diseases in patients were not analyzed; instead, the data from all patients were used [18,32,35]. First, the inclusion of only the initial ICU admission and exclusion of all subsequent ICU readmissions ensured that the outcome was measured the same way for all patients. This highlighted the early predictive ability of the model and prevented possible information omissions when the dataset was separated for training and testing (12,456 admissions were deleted). Second, the subjects used in this study were all adults older than 16 years (7878 admissions were deleted). Lastly, only data from patients who stayed in the ICU for longer than 24 h were utilized (2138 admissions were deleted). In the case of patients who stayed in the ICU for at least one day, only data from their first day were considered. If multiple measurements had been taken on the same day, an average of the values was taken. In addition, the data preprocessing method of Guo et al. [36] was used for the processing of missing values in this study. Three-stage missing value processing was carried out and patients with more than 30% missing variable values were excluded (8954 admissions were deleted) and a total of 27,550 participants were included in this study. Figure 2 depicts the extraction of data in their entirety.

In this study, information from the MIMIC-III database admission and chart events tables were used for the variable selection part. In reference to previous related studies [16,17,18,37], only six basic vital signs from the patient files were used: Heart Rate, Respiratory Rate, Systolic Blood Pressure, Diastolic Blood Pressure, Temperature, and Oxygen saturation, along with Glasgow Coma Scale and the patient’s age at admission as a predictor of variables in the first part. Topic model variables extracted from unstructured data of the initial diagnosis made by physicians when the patients were admitted to the hospital, were among the predictive variables in the second part.

### 2.3. Baseline Characteristics

Ultimately, the ICU records of 27,550 patients were utilized after the data in the MIMIC-III database had been preprocessed. Table 2 shows patient demographic information. The average age of the patients in this study was 64, of which 56% were male, their average hospital stay was 10.39 days, and their average intensive care unit stay was 4.48 days. In addition, over 84% of patients were admitted to the hospital for emergency care. Medicare insurance covered more than 50% of patients. Table 2 also displays the statistical values of the eight structural variables utilized in the study.

Among these were the diagnosis in the initial clinical notes about the patient made by the physician. As shown in Table 3, the diagnosis field provides the clinician with a written record of the initial diagnosis on admission. The admitting clinician usually specifies a diagnosis and does not use system ontology. Diagnoses may be very useful (e.g., congestive heart failure\biventricular implantable cardioverter defibrillator placement) or extremely vague (e.g., fever). This text section can provide useful information about the condition of the patient on admission. The information in the diagnosis field from the Admission Table was used in this study and a machine learning model was used to investigate the impact of structured EHR data and unstructured data on ICU patient mortality. Structured EHR data included variables such as vital signs and lab tests, and clinical note content includes topic features extracted from clinical notes using the LDA method.

### 2.4. Latent Dirichlet Allocation

Latent Dirichlet Allocation (LDA) is an unsupervised topic modeling algorithm that derives topics in a corpus. The model is a standard “bag of words” model, wherein each text item is viewed as a word frequency vector and the text is viewed as a set made up of various word groups [30]. Typically, an LDA topic generation model is built in three steps: First, a topic is extracted from the topic distribution for each text item. Second, a vocabulary corresponding to the extracted topics is taken from the vocabulary distribution. The steps are then repeated until every word in the text has been extracted. Because each text item contains multiple topics, several corresponding key words can be chosen for each topic. In other words, the same vocabulary can appear across multiple topics. Topic modeling methods mine significant topics from collected documents using probabilistic procedures and applications. As a result, by effectively processing a large amount of unstructured data in the text, the topic modeling method can help identify the latent semantics of complex articles [38,39]. LDA assumes that each document in the collection is created in two steps, the first by selecting a distribution of topics for that document, and the second by assigning a random topic and its corresponding distribution of words to each position in the document that may contain a word. This is repeated for the entire corpus. As a result, the main feature of LDA is that all documents share the same topic to varying degrees. Based on this theory, an LDA model can be applied to a set of documents using the Gibbs sampling algorithm to infer their underlying topics. The algorithm iterates over all the words in the document and calculates the most representative words for each topic. Each word can appear multiple times in the same document and can be repeated in different documents at the same time. At each iteration, the algorithm can modify the topic that best represents it, and after using Gibbs sampling with the training set, a model is built that produces a topic distribution for each document [40].

In a similar context-based textual analysis, probabilistic topic modeling conceptualizes a document as a collection of words derived from underlying thematic topics that define a probability distribution of words related to a topic, where the relative importance of each word in respective topics is defined by the conditional probability $P\left({\mathrm{Word}}_{i}{|\mathrm{Topic}}_{j}\right)$ in category probability distribution. Because an article is a weighted mixture of multiple topics, its conditional probability can be determined, and file content is generated based on the proportion of words related to each topic. This matrix is decomposed by topic modeling approaches based on latent topic structures that link latent words to related documents. A precise solution to this inverted inference is not generally tractable and requires an iterative optimization solution such as that given by Gibbs sampling. The probabilistic LDA framework will interpret correlation structures as conditional probabilities $P\left({\mathrm{Word}}_{i}{|\mathrm{Topic}}_{j}\right)$ and $P\left({\mathrm{Topic}}_{j}{|\mathrm{Document}}_{k}\right)$, which are closely related to other dimensionality reduction techniques for providing low-rank data approximations. An insight into the underlying topic structure allows for a more convenient, efficient, and interpretable approach to information retrieval, classification, and document data exploration [41].
(1)$$P\left({\mathrm{Word}}_{i}{|\mathrm{Document}}_{k}\right)=\sum_{j=1}^{J}P\left({\mathrm{Word}}_{i}{|\mathrm{Topic}}_{j}\right)\times P\left({\mathrm{Topic}}_{j}{|\mathrm{Document}}_{k}\right)$$

In the medical field, topic modeling research primarily focuses on the organization of clinical text, such as in newspapers and scientific literature, as well as clinical discharge records. However, recent studies have modeled laboratory results, claims data, and clinical concepts [42,43]. In this study, the aim is to learn the topic structure of clinical data through algorithms and apply it to clinical decision-making prediction. Unlike a top-down rule-based approach that isolates preconceived clinical concepts from electronic medical records, this bottom-up approach recognizes patterns in the data with more consistency. Additionally, in this paper, reference is made to an algorithm used in a previous study for the handling of non-quantified data [44,45], where Grid Search is used to confirm the best LDA model and tests multiple sets of topics. The LDA model was then applied to the ten topics derived from the results to categorize these key words into different topics. Any word that appears in a keyword set is related to the topic. Furthermore, certain words are more likely to appear under each topic, and there is a probability that each word will appear under respective topics. Figure 3 and Table 4 show the ten topics and keywords chosen for topic modeling in this study.

### 2.5. Machine Learning

In this study, the organized dataset was divided into two parts with 80% of the data being used for training the model and the remaining 20% for testing. Eight commonly used machine learning algorithms were used to establish the ICU mortality prediction model: Adaptive Boosting (AdaBoost), Bagging, Gradient Boosting, Light Gradient Boosting Machine (LightGBM), Logistic Regression, Multilayer Perceptron (MLP), Support Vector Classification (SVC), eXtreme Gradient Boosting (XGBoost). All data mining tasks of this research were conducted using the Python programming language. Table 5 shows the 8 machine learning models with their specific parameters’ settings. The following sections provide detailed descriptions of the various machine learning classification algorithms.

AdaBoost is an adaptive method in the sense that incorrect samples from the previous classifier are used to train the next classifier. The AdaBoost method is sensitive to noise and abnormal data. It trains a basic classifier and gives misclassified samples more weight. It is then applied to the next process. This iterative process is repeated until a stopping condition is reached or the error rate is low enough [46,47]. The Python sklearn library was used to implement AdaBoost. Our hyperparameters specified a maximum number of iterations of 50, while others trained the model using the sklearn preset values.The Bootstrap Aggregating algorithm, also known as the Bagging algorithm, is an ensemble learning algorithm in the field of machine learning, which was first proposed by Leo Breiman in 1994. The Bagging algorithm can be combined with other classification and regression algorithms to improve accuracy and stability while reducing result variance to avoid overfitting. Bagging is an ensemble method that combines multiple predictors. It helps to prevent model overfitting to data and reduces variance. It has been used in many microarray studies [48,49]. The Python sklearn library was also used to implement bagging. A combined classifier made up of 500 DecisionTreeClassifiers was used. Each classifier has a maximum sampling subset of 100; the self-service sampling method was used for each sampling. Other hyperparameters used sklearn preset values to carry out training.Gradient Boosting is an ensemble learning algorithm that can be used to improve the accuracy of various classification prediction models. It trains a model with poor prediction accuracy using the negative gradient information of the model loss function and then combines the trained results with the existing model in a cumulative form [50]. The scikit-learn library was also used in this study to achieve gradient boosting; the maximum number of iterations was set to 100; and other hyperparameters were trained using preset scikit-learn library values.The Light Gradient Boosting Machine (LightGBM) is an ensemble method that combines the predictions of multiple decision trees to produce a well-generalized final prediction. LightGBM divides continuous eigenvalues into K intervals and chooses dividing points from those intervals. This method significantly accelerates prediction and reduces memory occupancy without sacrificing prediction accuracy [51]. LightGBM is a decision tree learning algorithm with gradient boosting that has been widely used for feature selection, classification, and regression [52].Logistic Regression is a logit model capable of testing statistical interactions and controlling multivariate confidence. It is most commonly used to investigate the risk relationship between disease and exposure [53,54]. In this study, the Python scikit-learn library was used to implement logistic regression, and the hyperparameter optimization method was the SAG linear convergence algorithm. It is a gradient descent method used specifically for large sample data.Multilayer Perceptron (MLP) is a feed-forward artificial neural network with a fixed number of computational units or neurons that are fully connected to the next layer [55]. A multilayer perceptron learns and predicts data using the principles of the human nervous system. MLPs are suitable for classifying and predicting tasks with different feature set implementations [56]. The neural network used in this study had five layers: an input layer, three hidden layers, and an output layer. Each hidden layer has 13 neuron nodes, the normalization parameter was set to 1e-5, relu was used as the activation function, and adam was used for training and weight optimization.The Support Vector Classifier (SVC) analyzes linear and nonlinear data for classification and regression. SVC aims to recognize categories by the creation of non-linear decision hyperplanes in a higher feature space [57]. SVC is resistant to data bias and variance and produces accurate predictions for binary or multiclass classifications. Additionally, SVC is robust, resists overfitting, and has exceptional generalization capabilities [58].eXtreme Gradient Boosting (XGBoost) is a scalable end-to-end tree boosting system that is an optimized implementation of the gradient boosting framework. It is remarkable in that it can handle missing data efficiently, is very flexible, and can build an assembly of weak prediction models into an accurate one [59]. It generates a series of decision trees during training, each building on the previous one to reduce the loss function gradient. Furthermore, a predictive model made up of multiple decision trees can be obtained. The XGBoost algorithm can deal with missing values by including a default orientation for missing values in each tree node and learning the best orientation from the data [60].

### 2.6. The Synthetic Minority Oversampling Technique (SMOTE)

When the class distribution is highly skewed, machine learning problems become unbalanced. Unbalanced classification problems are prevalent in a variety of application domains and pose challenges for conventional learning algorithms [61]. In general, an imbalanced dataset can negatively affect the results of a model. In general, gold-standard datasets are unbalanced, which reduces model predictive ability [62]. In the evaluation of model performance, over- and underfitting are the most common issues. When a model has a high accuracy score during training but a low accuracy one during verification, overfitting has occurred. The greatest reduction in model overfitting can be achieved by increasing the size of the training set and decreasing the number of neural network layers. The failure of a model to classify data or make predictions during the training phase signifies underfitting [63]. SMOTE is a potent classification imbalance solution that produces consistent results across domains. The SMOTE algorithm adds synthetic data to the minority class to create a balanced dataset [61]. Class imbalance refers to the disparity between the classes of data used to train a predictive model, a prevalent issue that is not exclusive to medical data. Classification algorithms have a tendency to favor the majority class when it has significantly fewer observations than the class with negative outcome. Predictive performance can be improved by the manipulation of data, algorithms, or both [64]. The methodology involves the under- and oversampling of larger and smaller samples.

Table 6 displays the descriptive statistics for the data used in this study. The data in the table indicate a significant imbalance between the ratio of patient survival and mortality. Because these unbalanced datasets frequently produce inaccurate model prediction [65], the addition of minority class samples, or the deletion of majority class samples, is frequently performed to correct this [15]. The Synthetic Minority Oversampling Technique (SMOTE) randomly generates new minority class samples from the nearest neighbor line connecting the minority class samples and the technique is extensively used to process skewed data [63,66]. In this study, SMOTE technology was used to increase the sample size for the side with fewer samples to balance the data [15]. This was necessary because the number of samples of patients dying in the ICU was much smaller than the number of samples of patients surviving. In other words, a synthetic minority sampling technique was used to preprocess extremely unbalanced datasets.

In this study, a range of SMOTE methods of varying percentages was used to examine a selection of cases. A fresh training dataset was produced based on the information in Table 6. Non-survivors’ samples were increased by a factor of eight or nine using the SMOTE technology on a dataset of patients who died within 30 days of admission, from 3028 patients to 24,224 patients. This increased the proportion of the minority group in the baseline dataset from 10.99% to 49.69%.

### 2.7. Performance Evaluation

To make a thorough comparison of the impact of the integration of structured and unstructured data on the prediction of mortality in ICU patients, in this study, five different metrics were chosen as evaluation tools for modeling. These included AUROC, Precision, Recall, F1-Score, and Accuracy. Appendix A shows the confusion matrix.

## 3. Results

### 3.1. Prediction of Mortality in ICU

The k-fold cross-validation method was used to assess the performance of the model after training. The dataset was initially divided into k sections, with each section containing instances of equal size. The final measure of performance was the average of all test results across all components. This method has the benefit of training and validating all instances of the entire dataset, resulting in more accurate predictions with less bias. However, it is computationally costly, and validation is time-consuming. The model was constructed using 10-fold cross-validation, which has been utilized in a number of healthcare and medical studies [67,68]. In this study, the patient’s mortality was predicted at 3 days, 30 days, and 365 days after admission based on data collected within 24 h of admission. AUROC, which compares the true-positive rate to the false-positive rate, is the most prevalent metric used to evaluate the performance of diagnostic tools. Table 7 lists the eight distinct machine learning methods employed in this study, as well as AUROC for the ICU mortality prediction task across three all time periods. Our AUROC findings revealed that the mortality rate in 3 days can exceed 80%, and in 30 days and 365 days can exceed 75%. The results indicated that the best AUROC is 88.20% in our research, and that could accurately predict patient death within 3 days at 24 h after admission. Compared with using structured quantitative data alone, adding unstructured data makes the model increase by 2–5% on average in AUROC, which is a great improvement in the prediction of mortality of patients in intensive care units. Figure 4 shows that ICU data can be used to predict 3-day mortality with better precision. This clearly shows that the model developed in this study can predict the vital status of patients with great precision. Gradient Boosting, as indicated by the data in the chart, is the best model for predicting ICU patient mortality across all time periods.

For a comprehensive understanding of the impact of unstructured data on the prediction of mortality in ICU patients, the prediction results of models using only structured data within 24 h of ICU patient admission were compared with those using both structured and unstructured data. As illustrated in Figure 4, the pertinent prediction results are sorted. Within 24 h of ICU patient admission, the ROC of model prediction results using both structured and unstructured data is greater than that predicted using only structured data across all time periods. Table 7 also demonstrates that Gradient Boosting has a higher AUROC than other machine learning algorithms, regardless of ICU patient mortality across all time periods. Moreover, the prediction accuracy of the model made using both structured and unstructured data, within 24 h of patient admission to the ICU, is generally higher than that of the model using structured data alone. Indeed, basic observations and judgments of the patient at the time are of reference value and will significantly influence the accuracy of constructed model predictions. Overall, this indicates that a model constructed using both structured and unstructured data from ICU patients after admission can predict early patient death after admission with considerable accuracy. By incorporating unstructured data as input, it is possible to gain access to higher-level concepts not present in physiological data.

We summarize the use of four different metrics (Precision, Recall, F1-Score, and Accuracy) for a more complete picture of the differences in prediction accuracy of models constructed using different machine learning methods in Appendix B, which also show an evaluation of the structured ICU patient basic vital signs within 24 h of admission. These basic observations and judgement depend on whether or not the model was made using unstructured data about patient condition collected at time of admission to the ICU to predict time of patient death. According to the data in the table, the results obtained by using both structured and unstructured data of ICU patients after admission (and the eight different machine learning methods) are slightly better at predicting ICU patient mortality than those using structured data alone under different evaluation metrics. Furthermore, XGBoost has the highest prediction accuracy (97.23%) of the algorithms used, followed by LightGBM (95.61%), and Bagging has the highest prediction recall (95.13%)

### 3.2. Feature Importance

The most promising features are typically chosen, and the unimportant ones are usually eliminated using feature selection methods. The feature importance score reflects the information gained by each feature during construction of the decision tree [69]. An advantage of using Gradient Boosting is that, once the prediction model has been constructed, the variable importance can be obtained with relative ease by sorting the calculated variable importance scores. The feature importance framework ranks input variables according to their contribution to the predictive model and gives insight into which features are crucial for the task [70]. The more a variable is utilized in the decision tree, the more important it will become. In this study, the importance of each feature is determined by applying a feature importance scoring method to a model trained with gradient boosting. In addition, a percentage rating is provided for how frequently each feature is used to determine the output label. Relevant research notes [71,72] provide additional information on how the Gradient Boosting method determines the significance of input variables. The variance importance within 24 h of ICU patient admission is outlined in Table 8.

According to Table 8, the Glasgow Coma Scale (X_7_), Age (X_8_), and Heart Rate (X_1_) are relatively important variables for prediction of ICU patient mortality using structural data from ICU patients recorded within 24 h of admission. The addition of initial clinical diagnosis records (unstructured data) produced variable results about the feature significance of patient mortality prediction. In addition to the Glasgow Coma Scale (X_7_), chest pain (TOPIC_6_) and coronary artery disease (TOPIC_1_) were also relatively significant. In the model constructed using data from ICU patients within 24 h of admission, bleed mass (TOPIC_7_) was a relatively important variable for 3-day mortality, altered mental status (TOPIC_10_) for 30-day mortality, and sepsis (TOPIC _8_) for 365-day mortality. Important variables to consider are the Glasgow Coma Scale (X_7_), chest pain (TOPIC_6_), and coronary artery disease (TOPIC_1_), regardless of the mortality prediction for different ICU patient time periods.

## 4. Discussion

### 4.1. Principal Findings

Previous studies have focused on building predictive models using quantitative variables from EHR databases to predict mortality, length of stay, and disease diagnosis in ICU patients. However, such studies have largely ignored the potential value of qualitative data due to challenges in standardization and utilization. By overlooking unstructured data in EHR, clinicians may miss critical information and clues provided by the physician’s initial observations. To fully utilize unstructured data, this study employs NLP techniques, specifically the LDA model, to analyze clinical notes. Our study integrates structured data, such as basic vital signs, with unstructured data, derived from physicians’ initial clinical diagnoses at the time of ICU admission, to predict patient mortality. Additionally, our model successfully identifies significant variables for predicting clinical outcomes during different ICU periods. We hope that our analysis results can enhance medical staff’s understanding of patient conditions, optimize medical resource allocation, and provide patients, families, and medical staff with more information for informed decision-making. The main contributions of this study include: (1) investigating the impact of integrating structured and unstructured clinical records on ICU patient outcomes using a machine learning model, and (2) predicting patient mortality and risk factors to inform potential preventive measures in medical practice.

In previous studies, researchers have achieved comparable or even superior accuracy by employing excessive numbers of features. For instance, Xia et al. [13] used 50 features to achieve an AUROC of 0.85, and Liu et al. [73] employed 99 features to achieve an AUROC of 0.78. However, in our study, we achieved an accuracy of 97% and an AUROC of 0.88 for the mortality model using only six vital signs, the GCS, age, and the initial written clinical records and diagnosis made on patient admission to the ICU. We utilized eight commonly used machine learning classification algorithms, each with a known degree of accuracy in predicting ICU patient mortality. Our AUROC findings revealed that the mortality rate in 3 days can exceed 80%, and in 30 days and 365 days can exceed 75%. Our study found that Gradient Boosting provided the most accurate prediction model. XGBoost had the highest prediction accuracy, indicating that our proposed method could predict mortality in ICU patients very well. Our results also demonstrated that the initial written notes of clinical observations and diagnoses made at the time of patient admission to the ICU contain a wealth of useful information that can aid ICU medical and nursing staff in making crucial clinical decisions. Furthermore, our study only utilized structured and unstructured data of ICU patients within 24 h of admission; our prediction model was found to be more suitable for predicting short-term mortality, as it could predict 3-day mortality with more accuracy than 30-day and 365-day mortality.

Using the LDA method, the analysis of unstructured data recorded by ICU admission clinicians during initial observation and diagnosis yielded significant results. Other important variables to consider in addition to the Glasgow Coma Scale (X_7_) are patient age at admission (X_8_), chest pain (TOPIC_6_) and coronary artery disease (TOPIC_1_). Overall, the LDA method can extract significant medical characteristics from patient topics. Furthermore, these medical characteristics can be utilized in a variety of situations to provide personalized clinical advice to individual patients [35]. In addition, various imputation techniques were applied to the dataset to determine the optimal solution for the issue at hand. Because the number of ICU patients who died in this study was significantly lower than the number of those who survived, the majority-to-minority ratio was 97 to 3 (3-day mortality) and the data demonstrate an extremely high category imbalance. SMOTE technology was used to increase the sample size of the side with the smaller number of samples to achieve data parity.

### 4.2. Limitations

To begin, all the data used in this paper came from a large retrospective clinical database, and the findings were generalized across groups of patients rather than specific people. To ensure the thorough collection of relevant data, this study only took into account complete patient dynamic information from databases where ICU data were easy to obtain. This study used MIMIC-III data collected at Beth Israel Deaconess Medical Center in Boston, Massachusetts. Future studies should evaluate data collected from more medical facilities across a wider geographic area. Because this study was limited to ICU medical data accumulated by a large medical facility in a big city, the findings cannot be safely applied to ICU patients in smaller medical facilities. More comprehensive results and verification could be obtained by comparing these results with those from data obtained from rural or other general small medical facilities.

Second, the model’s performance may be undermined in other critical care settings due to a lack of high-quality care notes for a large number of patients. The information entered by physicians during patient consultations is valuable for disease and treatment research. Because these notes are highly telegraphic and contain many spelling errors, inconsistent punctuation, and non-standard word order, the existing natural language analysis tools struggle to process them [74]. Common spelling errors and other noise in medical notes can affect interpretation quality, resulting in counterintuitive results, which is a limitation and challenge for related research [5]. Furthermore, because this study is retrospective, conclusions about predictive algorithm performance in a hospital setting cannot be drawn. Future studies could evaluate and analyze these constraints in greater depth to make this kind of study more objective and thorough.

## 5. Conclusions

As the COVID-19 pandemic continues to strain ICUs worldwide, the critical importance of such facilities has become increasingly apparent. Consequently, there is a pressing need for more active research to manage scarce critical care resources and provide additional tools to support medical decision-making and effective benchmarks for clinical practice. In this study, not only using structured data from ICU patients’ first 24 h (including six vital sign measurements, the GCS, and patient age at admission), but also focusing on unstructured data from initial state observations and diagnoses made upon admission. The effectiveness of using LDA method and different machine learning technologies in the prediction of ICU patient mortality was discussed. These unstructured data contained a wealth of information that could effectively assist in later clinical decision making. However, the model developed in this study primarily focused on predicting ICU patient mortality, and further investigation is warranted to explore other clinical tasks such as length of stay, complication, and disease prediction. Moreover, it is evident that physician-produced clinical care records may capture the concepts required for mortality prediction with greater pertinence and accuracy than is currently achievable using traditional statistical techniques. Therefore, it is recommended that four directions be pursued for further research.

First, clinicians should integrate a large amount of information to evaluate and predict the current and future status of the patient, making the environment of critical care cognitively more demanding. It is essential to gain a comprehensive understanding of the specific need for clinical ICU predictive systems, the types and properties of predictions that are valued by the clinician, and the optimal time scale for such predictions. Despite the fact that findings show that our proposed method produced good predictive results for ICU patient mortality, additional research is required to evaluate its benefits on clinical care and its effectiveness to elucidate the prediction principles.

Second, the data in this study were restricted to patients admitted to the ICU for the first time and exclude patient readmission records and reports. Reduction in readmissions has long been identified by the United States government as a priority area for healthcare policy reform. Hospital readmission has also been promoted as a metric that can aid in the reduction in healthcare cost. More types of readmission research, such as the predictive performance of readmission models, could be conducted, as well as the impact of patient-level predictors on readmission, and studies of the relationship between healthcare environment quality and readmission [75,76]. Because ICU patient readmission frequently results in excessive use of medical resources and financial risk to medical facilities, analyzing the morbidity and mortality of readmitted ICU patients will benefit both patients and medical facilities [77]. Future research could collect data from multiple ICU admissions and make a comprehensive evaluation of time-series issues and also provide additional levels of analytical results as a reference for patients, medical and nursing staff, and the families of the patients.

Third, because the MIMIC-III database contains accumulated medical data from ICU patients at the medical facilities of a large city, the results of the analysis cannot be safely applied to ICU patients at smaller medical facilities. If follow-up studies are made using ICU patient data from rural and other general medical facilities as a comparison, more comprehensive results and verification would be available. Patient data should be collected from different medical centers, including outpatient, inpatient, and emergency facilities, as well as ICUs. This will allow a more comprehensive model to be constructed for evaluation and expand applicability. In addition, classification and analysis could be conducted on the basis of various diseases, such as diabetes, and disciplines such as chest medicine.

Finally, unstructured or semi-structured data account for more than 80% of the information in electronic health records. If qualitative information is ignored, clues and key factors may be missed if the initial observation-based judgments of the physician are not taken into account. In this study, unstructured data from the initial state observation and diagnosis made by physicians at ICU admission were added to the commonly used structural variables in the traditional ICU prediction model and the LDA method was used for model construction. Future research can also collect and integrate various types of unstructured data, such as the hospital consultation process, the needs of the patient, and their social media message content, to improve prediction accuracy of the model. Other new topic modeling tools, such as BERT, can also be used to assess the power of the proposed prediction plan.

## Figures and Tables

**Figure 1 ijerph-20-04340-f001:**
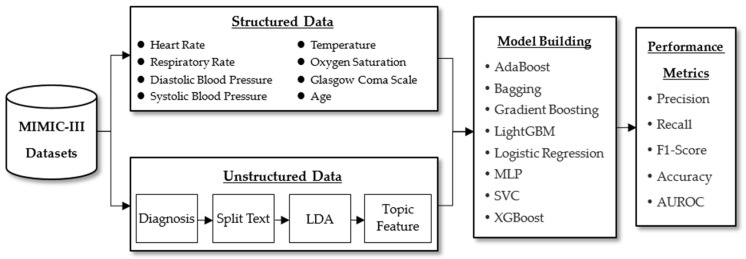
Research scheme.

**Figure 2 ijerph-20-04340-f002:**
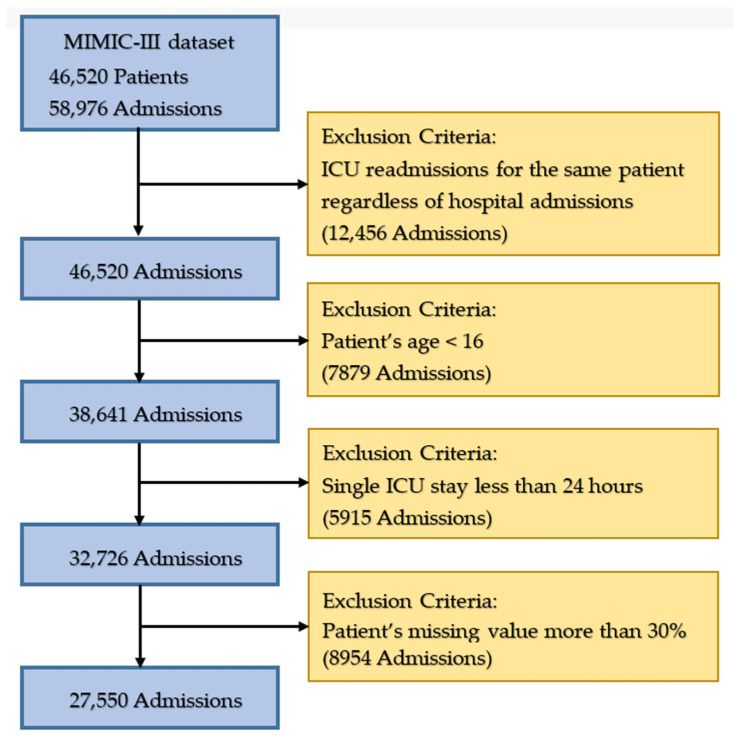
The process of data extraction.

**Figure 3 ijerph-20-04340-f003:**
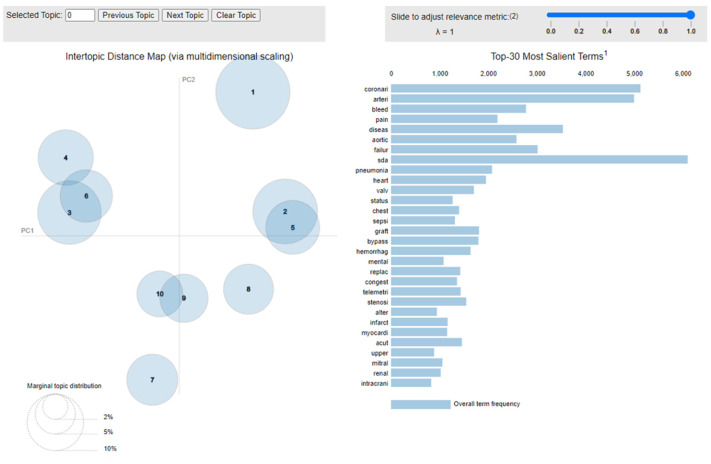
Topic modelling in Python.

**Figure 4 ijerph-20-04340-f004:**
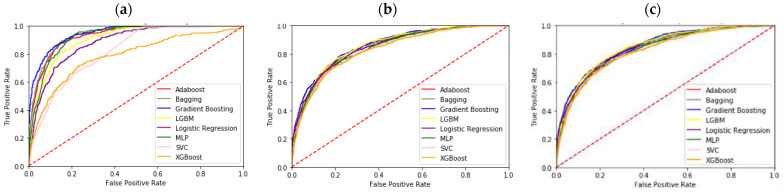
ROC curves for the different classifiers. (**a**) 3-day mortality. (**b**) 30-day mortality. (**c**) 365-day mortality.

**Table 1 ijerph-20-04340-t001:** The summary of MIMIC-III dataset.

Distinct Patients	46,520
Age, years, median [Q1–Q3]	65.8 [52.8–77.8]
Gender, male, percent of unit stays	26,121 (56.15%)
Distinct hospital admissions	58,976
Elective	7706 (13.07%)
Emergency	42,071 (71.34%)
Newborn	7863 (13.33%)
Urgent	1336 (2.27%)
Hospital mortality, percent of unit stays	5854 (9.93%)
Hospital length of stay, median days [Q1–Q3]	10.13 [3.74–11.80]
Distinct ICU stays	61,532
Coronary Care Unit	7726 (12.56%)
Cardiac Surgery Recovery Unit	9312 (15.13%)
Medical Intensive Care Unit	21,088 (34.27%)
Neonatal Intensive Care Unit	8100 (13.16%)
Surgical Intensive Care Unit	8891 (14.45%)
Trauma Surgical Intensive Care Unit	6415 (10.43%)
ICU length of stay, median days [Q1–Q3]	4.92 [1.11–4.48]

**Table 2 ijerph-20-04340-t002:** Features involved in the model.

	Total	Survivors	Non-Survivors
General			
Number	27,550 (100%)	24,364 (88.44%)	3186 (11.56%)
Gender (male)	15,441 (56.05%)	13,764 (89.14%)	1677 (10.86%)
Length of stay			
Hospital (days) [Q1–Q3]	10.39 [4.36–12.42]	10.29 [4.44–12.35]	11.20 [3.88–12.54]
ICU (days) [Q1–Q3]	4.48 [1.55–4.61]	4.18 [1.53–4.33]	6.77 [1.70–6.13]
Admission Type			
Elective	3537 (12.84%)	3434 (14.09%)	103 (3.23%)
Emergency	23,283 (84.51%)	20,293 (83.29%)	2990 (93.85%)
Urgent	730 (2.65%)	637 (2.61%)	93 (2.92%)
Insurance			
Government	844 (3.06%)	799 (3.28%)	45 (1.41%)
Medicaid	2301 (8.35%)	2078 (8.52%)	223 (7.00%)
Medicare	14,750 (53.54%)	12,618 (51.79%)	2132 (66.92%)
Private	9303 (33.77%)	8570 (35.17%)	733 (23.01%)
Self-Pay	352 (1.28%)	300 (1.23%)	52 (1.63%)
Variable value (First 24 h)			
Heart Rate	85.72 ± 15.91	85.11 ± 15.52	90.38 ± 17.93
Respiratory Rate	18.94 ± 4.01	18.70 ± 3.84	20.75 ± 4.76
Diastolic Blood Pressure	60.33 ± 12.21	60.66 ± 12.09	57.79 ± 12.79
Systolic Blood Pressure	118.01 ± 18.60	118.45 ± 18.30	114.63 ± 20.45
Temperature	98.23 ± 2.03	98.28 ± 1.73	97.90 ± 3.54
Oxygen Saturation	97.21 ± 2.11	97.28 ± 1.90	96.68 ± 3.28
Glasgow Coma Scale	12.31 ± 3.21	12.66 ± 2.91	9.64 ± 4.00
Age	64.00 ± 17.71	63.13 ± 17.75	70.66 ± 15.81

**Table 3 ijerph-20-04340-t003:** Patients’ diagnosis records.

SUBJECT_ID	HADM_ID	Diagnosis
00412	109897	AORTIC STENOSIS; MITRAL REGURGITATION; CAD\AORTIC VALVE REPLACEMENT; MITRAL VALVE REPLACEMENT; CORONARY ARTERY BYPASS GRAFT; TRICUSPID VALVE REPLACEMENT/SDA
00969	137250	BATTERY DEPLETION; HEART FAILURE\IMPLANTABLE CARDIOVERTER DEFIBRILLATOR EXPLANT; PACEMAKER IMPLANT; DIURISIS POST PROCEDURE/SDA
14229	145873	MARFAN’S SYNDROME\BENTALL PROCEDURE; TOTAL VALVE SPARING ROOT REPLACEMENT VS; HOMOGRAFT ROOT REPLACEMENT; REPLACEMENT OF ARCH, PROXIMAL ROOT/SDA
22416	130625	DESCENDING AORTIC ANEURYSM; COARCTATION OF DESCENDING AORTA\ DISTAL ARCH REPLACEMENT; DESCENDING THORACIC AORTIC REPLACEMENT; AORTA TO SUBCLAVIAN BYPASS/SDA
23360	104836	POLYCHONDRITIS WITH AIRWAY MANISFESTATION\ STERNATOMY CARDIOPULMONARY; BYPASS; ANTERIOR TRACHEAL SPLITTING; TY STENT PLACEMENT; LAPAROTOMY/SDA
28352	154475	PULMONARY VEIN INJURY\THORACOSCOPIC MAZE PROCEDURE LEFT; MINI MAZE; BILATERAL MINI THORACOTOMY; PULMONARY VEIN ISOLATION; RESECTION OF LEFT ATRIAL APPENDAGE/SDA
45688	144761	RIGHT VENTRICULAR LEAD MALFUNCTION; INAPPROPRIATE IMPLANTABLE CARDIOVERTER-DEFIBRILLATOR FRING\RIGHT VENTRICULAR IMPLANTABLE CARDIOVERTER-DEFIBRILLATOR LEAD EXTRACTION/SDA
51821	182983	MEDIASTINAL ADENOPATHY\FLEXIBLE BRONCHOSCOPY; LINEAR ENDOBRONCHIAL ULTRASOUND (EBUS); FLUOROSCOPY; TRANSBRONCHIAL BIOPSY; TRANSBRONCHIAL NEEDLE ASPIRATION; BRONCHIAL ALVEOLAR LAVARGE
92284	193856	AIRWAY OBSTRUCTION\FLEXIBLE BRONCHOSCOPY; RADIAL ENDOBRONCHIAL ULTRASOUND (EBUS); BRONCHIAL AVEOLAR LAVAGE/ BRUSH; POSSIBLE TRANSBRONCHIAL BIOPSY (LEFT UPPER LOBE); FLUOROSCOPY

**Table 4 ijerph-20-04340-t004:** Topics and keywords for dataset.

Variable	Topic	Keywords
TOPIC_1_	Coronary Artery Disease	coronari, arteri, diseas, graft, bypass, sda, syndrom, effus, cath, avr, acut, etoh, pericardi, cerebr, pleural, cholang, mvr, leav, vascular, angioplasty.
TOPIC_2_	Aortic Valve Replacement	aortic, sda, valv, replac, stenosi, mitral, cancer, subarachnoid, hemorrhag, esophag, procedur, ascend, regurgit, aorta, maze, airway, redo, bental, repair, invas, obstruct
TOPIC_3_	Heart Failure	failur, heart, congest, acut, infarct, myocardi, renal, cath, liver, pancreat, elev, cardiac, dehydr, rule, cholecyst, hyperkalemia, leukemia, lacer, hyperglycemia, block, chronic, implant
TOPIC_4_	Pneumonia	pneumonia, telemetri, fractur, stroke, ischem, atrial, attack, transient, angina, dyspnea, fibril, hip, unstabl, cath, chronic, segment, diseas, pulm, obst, ablat, cardiomyopathi, pelvic, septal
TOPIC_5_	SDA	sda, right, aneurysm, leav, accid, motor, vehicl, short, breath, tachycardia, cellul, ventricular, abdomin, lung, injuri, hepat, bilater, metastat, perfor, colon, spinal
TOPIC_6_	Chest Pain	pain, chest, hemorrhag, intracrani, fever, abdomin, hypotens, fall, cath, telemetri, dissect, insuffici, stroke, cardiac, femur, strike, epidur, neck, pedestrian, skull, cervic
TOPIC_7_	Bleed Mass	bleed, upper, lower, mass, pulmonari, obstruct, bowel, head, brain, weak, emboli, bradycardia, hypertens, stemi, small, edema, hemoptysi, cirrhosi, vomit
TOPIC_8_	Sepsis	sepsi, infect, asthma, exacerb, copd, sda, urinari, tract, tumor, brain, catheter, overdos, leav, anemia, pyelonephr, syncop, bscess, foot, ulcer, disord
TOPIC_9_	Subdural Hematoma	hematoma, subdur, respiratori, diabet, seizur, ketoacidosi, trauma, failur, blunt, sda, withdraw, distress, hyponatremia, wind, hernia, remot
TOPIC_10_	Altered Mental Status	status, mental, alter, arrest, cardiac, carotid, hypoxia, chang, leg, angiogram, stenosi, transplant, kidney, chf, accid, extrem, cerebrovascular, fib, stent, thrombosi, ischemia

**Table 5 ijerph-20-04340-t005:** Machine learning models with their specific parameters’ settings.

Model	Parameters
AdaBoost	*base_estimator = DecistionTreeClassifer, random_state = 1, n_estimators=50, learning_rate = 1.0, algorithm = ‘SAMME.R’*
Bagging	*base_estimator = None, n_estimators = 500, max_samples = 100, bootstrap = True, bootstrap_features = False, oob_score = False, warm_start = False, n_jobs = 1, random_state = None, verbose = 0*
Gradient Boosting	*n_estimators = 100, learning_rate = 1.0, max_depth = 1, random_state = 0*
LightGBM	*boosting_type = ‘gbdt’, num_leaves = 31, max_depth = −1, learning_rate = 0.1, n_estimators = 100, subsample_for_bin = 200,000, min_child_samples = 20, subsample = 1.0, subsample_freq = 0, colsample_bytree = 1.0, reg_alpha = 0.0, reg_lambda = 0.0 n_jobs = −1, importance_type = ‘split’,*
Logistic Regression	*solver = ‘sag’, penalty = ’l2′, max_iter = ’max_iter’*
MLP	*solver = ‘adam’, alpha = 1e-5, hidden_layer_sizes = (13,13,13), max_iter = 1000*
SVC	*C = 1.0, kernel = ‘rbf’, degree = 3, gamma = ‘auto’, coef0 = 0.0, shrinking = True, probability = False, tol = 0.001, cache_size = 200, class_weight = None, verbose = False, max_iter = −1*
XGBoost	*n_estimators = 100, booster = ’gbtree’, eta = 0.3, min_child_weight = 1, max_depth = 3, gamma = 0, max_delta_step = 0, subsample = 1, colsample_bytree = 1, colsample_byleve = 1, lambda = 1, learning_rate = 0.1, n_jobs = 1, base_score = 0.5, max_delta_step = 0, min_child_weight = 1*

**Table 6 ijerph-20-04340-t006:** Demographic information and SMOTE technique of the selected patient cohort.

	3 Days	30 Days	365 Days
Number of patients	27,550	27,550	27,550
Number of survivors	26,640	24,522	24,364
Number of non-survivors	910	3028	3186
Mortality ratio	3.30%	10.99%	11.56%
SMOTE increase	2900%	900%	900%
Number of survivors	26,640	24,522	24,364
Number of non-survivors	26,390	24,224	22,302
Mortality ratio	49.76%	49.69%	47.79%

**Table 7 ijerph-20-04340-t007:** AUROC of different classifiers.

		3 Days	30 Days	365 Days
Structured Data	AdaBoost	0.8530 ± 0.0041	0.7514 ± 0.0095	0.7478 ± 0.0058
	Bagging	0.8568 ± 0.0073	0.7627 ± 0.0054	0.7526 ± 0.0049
	Gradient Boosting	0.8598 ± 0.0082	0.7634 ± 0.0126	0.7588 ± 0.0053
	LightGBM	0.8159 ± 0.0149	0.7594 ± 0.0062	0.7523 ± 0.0035
	Logistic Regression	0.8110 ± 0.0221	0.7396 ± 0.0076	0.7353 ± 0.0063
	MLP	0.8494 ± 0.0163	0.7571 ± 0.0097	0.7493 ± 0.0060
	SVC	0.8097 ± 0.0040	0.7487 ± 0.0103	0.7443 ± 0.0056
	XGBoost	0.7070 ± 0.0115	0.7215 ± 0.0070	0.7201 ± 0.0041
Structured Data + Unstructured Data	AdaBoost	0.8686 ± 0.0076	0.7531 ± 0.0066	0.7629 ± 0.0114
	Bagging	0.8713 ± 0.0059	0.7644 ± 0.0098	0.7725 ± 0.0048
	Gradient Boosting	0.8820 ± 0.0119	0.7815 ± 0.0073	0.7754 ± 0.0091
	LightGBM	0.8361 ± 0.0143	0.7780 ± 0.0118	0.7705 ± 0.0036
	Logistic Regression	0.8298 ± 0.0109	0.7618 ± 0.0102	0.7502 ± 0.0051
	MLP	0.8679 ± 0.0168	0.7693 ± 0.0112	0.7540 ± 0.0042
	SVC	0.8142 ± 0.0082	0.7518 ± 0.0110	0.7512 ± 0.0040
	XGBoost	0.7655 ± 0.0174	0.7379 ± 0.0067	0.7345 ± 0.0044

**Table 8 ijerph-20-04340-t008:** The important variables by using Gradient Boosting.

Dataset	Variable Importance	3 Days	30 Days	365 Days
Structured Data	1	X_7_	X_7_	X_7_
	2	X_8_	X_8_	X_8_
	3	X_1_	X_1_	X_1_
	4	X_4_	X_4_	X_4_
	5	X_3_	X_5_	X_5_
Structured Data + Unstructured Data	1	TOPIC_6_	X_7_	X_7_
	2	X_7_	TOPIC_1_	TOPIC_1_
	3	TOPIC_1_	TOPIC_6_	TOPIC_6_
	4	TOPIC_7_	X_8_	X_8_
	5	TOPIC_8_	TOPIC_10_	TOPIC_8_

## Data Availability

Not applicable.

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
