# Peer review of "Integrating Structured and Unstructured EHR Data for Predicting Mortality by Machine Learning and Latent Dirichlet Allocation Method"

_ijerph, 2023, doi:10.3390/ijerph20054340_

Round 1

Reviewer 1 Report

In general, this is an interesting attempt to study the factors to predict motility in ICUs via machine learning approaches. I would recommend "accept with minor revisions". The following concerns for current version are listed for reference.

1.      The manuscript used several big paragraphs ie line 204 to line 244, line 265 to line 335, line 386 to line 404 to introduce basic methodologies or common metrics, which could be revised or placed in the appendix 

2.      From all the experiments completed in the study, The data incorporated unstructured data improved the performance of prediction consistently in all the models via various metrics. It is also true for different time horizons. According to Table 12, unstructured data play more important roles in the features extracted by Gradient Boosting, ie Topic 1 and Topic 6. For comparison only, the authors can conduct a test with unstructured data only to see their prediction power.

3.      The manuscript need improve a bit due to some grammar issues.  

Author Response

Response to Reviewer #1

We would like to thank you for your valuable recommendations that make the paper more complete and readable. We have made the following adjustments based on your suggestions.

In general, this is an interesting attempt to study the factors to predict motility in ICUs via machine learning approaches. I would recommend "accept with minor revisions". The following concerns for current version are listed for reference.

  1. The manuscript used several big paragraphs ie line 204 to line 244, line 265 to line 335, line 386 to line 404 to introduce basic methodologies or common metrics, which could be revised or placed in the appendix 

Corrections and adjustments:

Thanks for your valuable suggestions. We have revised and replaced our manuscript especially in method and result. Thank you so much.

  1. From all the experiments completed in the study, the data incorporated unstructured data improved the performance of prediction consistently in all the models via various metrics. It is also true for different time horizons. According to Table 12, unstructured data play more important roles in the features extracted by Gradient Boosting, ie Topic 1 and Topic 6. For comparison only, the authors can conduct a test with unstructured data only to see their prediction power.

Corrections and adjustments:

Thank you very much for reading our article so carefully. In our previous publication [1], the analysis of patients admitted to the ICU included unstructured data from the initial clinical diagnosis made by the patient's physician at the time of admission. Due to the consideration of the attributes of medical text reports, it often happens that the NOTES of medical staff only adjusts the numbers, and the content is too repetitive.

Therefore, this research we only used the physician's first diagnosis record at the time of patient admission. And we use basic vital signs in the selection of structured data. Among these were the diagnosis in the initial clinical notes about the patient made by the physician. The diagnosis field provides the clinician with a written record of the initial diagnosis on admission. The admitting clinician usually specifies a diagnosis and does not use system ontology. Diagnoses may be very useful (e.g., congestive heart failure \ biventricular implantable cardioverter defibrillator placement) or extremely vague (e.g., fever). This text section can provide useful information about the condition of the patient on admission.

However, modeling tests using unstructured data alone did not give good results due to the limited content of the initial diagnosis transcript. In addition, our research tried to set the LDA classification as 5/10/15/20 topics, and the results obtained were 10 topics as the best. This is the result presented in our article now. The reviewer's suggestion is to simply use topics with a higher ranking (reduce the number of topics, and the keywords of each topic will be reorganized). Thank you very much for your suggestions, which are important nutrients and support for our future research and innovation

  1. The manuscript need improve a bit due to some grammar issues.  

Corrections and adjustments:

Thanks for your valuable suggestion. We have optimized and modified the article again, looking forward to a more complete and easy-to-understand presentation. Once again, we would like to thank the reviewer for carefully reading our article and giving us encouragement and suggestions.

Thank you again for taking the time out of your busy schedule to provide valuable suggestions for our articles. These suggestions lead us to revisit the original article for inaccuracies and completeness. Because of your suggestions, the article became more complete, clear and readable. Thank you so much.

  1. Chiu, C.-C., et al. Predicting the Mortality of ICU Patients by Topic Model with Machine-Learning Techniques. in Healthcare. 2022. MDPI.

Reviewer 2 Report

1. Write full-term at the first occurrence of the MIMIC abbreviation.

2. MIMIC-III is free, publicly available multi-year critical care hospitalization data. Please add a specific description of MIMIC-III along with why it is suitable for this study.

3. Add the result of classifying the principal diagnosis of the patient used in the study using the International Classification of Diseases standard code. It is very important to present this because the performance of this study differs depending on the principal diagnosis distribution.

4. Age was analyzed in duplicate in Table 2. Delete either one.

5. Too many tables. Reduce the number of tables for readability. Please select the indicators that are mainly covered in the research results and integrate Tables 9~11 into one table.

6. When writing research results, they should be described objectively, including important numbers.

7. Discussion should be described while comparing with various previous literature, focusing on the research achievements shown in the research results. There is a lack of previous literature cited in the discussion(principal findings). Focusing on important variables in predicting mortality, please write a rich discussion by comparing it with a wide range of previous studies.

Author Response

Response to Reviewer #2

We would like to thank you for your valuable recommendations that make the paper more complete and readable. We have made the following adjustments based on your suggestions.

  1. Write full-term at the first occurrence of the MIMIC abbreviation.

Corrections and adjustments:

Thanks for your valuable suggestions. As the reviewer pointed out, we have added the full-term at the first occurrence, Medical Information Mart for Intensive Care (MIMIC). Thank you so much.

  1. MIMIC-III is free, publicly available multi-year critical care hospitalization data. Please add a specific description of MIMIC-III along with why it is suitable for this study.

Corrections and adjustments:

Thanks for your valuable suggestions. We have added more description of MIMIC-III.

The digital health system has developed rapidly in recent years and has been widely used by major hospitals. Even so, due to some reasons such as personal privacy and security, it is difficult to integrate and apply this information to scientific research. The clinical data performance of MIMIC-III was satisfactory in terms of data completeness, accessibility and model prediction completeness. The open use of MIMIC-III has been widely used in scientific research fields, contributing to the research and development of patient prognosis prediction and entity recognition. In order to consider the convenience and completeness of data collection, this study only considers obtaining complete patient dynamic information from databases that are easier to obtain ICU data.

  1. Add the result of classifying the principal diagnosis of the patient used in the study using the International Classification of Diseases standard code. It is very important to present this because the performance of this study differs depending on the principal diagnosis distribution.

Corrections and adjustments:

Added the results of classifying the initial diagnoses of the patients used in the study using the International Standard Classification of Diseases codes. This is a great suggestion. In many studies in the past, academics joined the study using ICD-9 or ICD-10 and got good results as a result.

The data we collected in this paper mainly focused on the initial data of admitted patients (including structured vital signs and unstructured written records of doctors' initial diagnosis). In the MIMIC-III database, records related to ICD-9 were recorded in the patients' discharge records. Nevertheless, we will add factors such as ICD-9 or ICD-10 to our data in future studies, expecting better results. Thanks to the reviewers for giving us such good suggestions.

  1. Age was analyzed in duplicate in Table 2. Delete either one.

Corrections and adjustments:

Thanks for your valuable suggestions. This is our negligence in our article. Thank you very much for your careful and detailed review, because with your support and reminders, we can have better articles.

  1. Too many tables. Reduce the number of tables for readability. Please select the indicators that are mainly covered in the research results and integrate Tables 9~11 into one table.

Corrections and adjustments:

Thanks for your valuable suggestions, we have moved Tables 9-11 to Appendix, which can indeed make the whole article more concise and improve readability. Thank you for giving us such a great suggestion.

  1. When writing research results, they should be described objectively, including important numbers.

Corrections and adjustments:

Thanks for your valuable suggestions. We have add some important numbers in our result that make the our article more complete and readable.. Thank you so much.

  1. Discussion should be described while comparing with various previous literature, focusing on the research achievements shown in the research results. There is a lack of previous literature cited in the discussion(principal findings). Focusing on important variables in predicting mortality, please write a rich discussion by comparing it with a wide range of previous studies.

Corrections and adjustments:

Thanks for your valuable suggestions. We have rework our conclusion and discussion. That’s very important to make our research more organized and make the article clearer and easier to read. Thank you so much.

Thank you again for taking the time out of your busy schedule to provide valuable suggestions for our articles. These suggestions lead us to revisit the original article for inaccuracies and completeness. Because of your suggestions, the article became more complete, clear and readable. Thank you so much.

Reviewer 3 Report

THe paper show an application of an intersting  approach to the use of a sort of free text to have further information to aply in a ML algorithm to predict outcome in ICU.

THe paper, in my opinio, is well written, even if more attention should be paid to english (small typing errors scattered in the paper).

HEre following suggestions and request to be clear in soma point of the paper.

Introduction.

THe first 67 rows of the introduction explain well the topic, but more details should be given for subsequent arguments (rows 67-70) and to the main aim of the paper (rows 118-124).

MEthods.

I expected to read the use of daily notes and other free text in the medical records. I think that it should be be better highlighted that unstructured data is limited only to admission ICU diagnosis) and that the model uses initial condition (structured and unstructured) only. A short explanation of "lenght of admission could be useful for those not near the ICU practice (row 143). THe rows 144-148 explain amount of data per patients per stay: this could be confusing respect to the use of initial data and should be better explained.

THe esclusion of readmission need a better clinical explanation of the choice.

In table 2 the statistics should be declared (median or mean, confidence interval or standard deviation).

I ask if the words in table 4 are truncated to take more cases or are typing errors.

if recall and sensistivity are the same it's better to call the measue the same way in table, formula and text; eventually in the text at rows 393 could be stated as Recall (i.e. sensitivity). 

DIScussion.

REsults and methods should be discussed more in depth.

Best regards

Author Response

Response to Reviewer #3

THe paper show an application of an intersting  approach to the use of a sort of free text to have further information to aply in a ML algorithm to predict outcome in ICU.

THe paper, in my opinio, is well written, even if more attention should be paid to english (small typing errors scattered in the paper).

HEre following suggestions and request to be clear in soma point of the paper.

We would like to thank you for your valuable recommendations that make the paper more complete and readable. We have made the following adjustments based on your suggestions.

  •  

THe first 67 rows of the introduction explain well the topic, but more details should be given for subsequent arguments (rows 67-70) and to the main aim of the paper (rows 118-124).

Corrections and adjustments:

Thanks for your valuable suggestions. As the reviewer pointed out, we have reworked for the subsequent arguments and the aim of our article that make the introduction more clear and readable. Thank you so much.

  •  

I expected to read the use of daily notes and other free text in the medical records. I think that it should be be better highlighted that unstructured data is limited only to admission ICU diagnosis) and that the model uses initial condition (structured and unstructured) only. A short explanation of "lenght of admission could be useful for those not near the ICU practice (row 143). THe rows 144-148 explain amount of data per patients per stay: this could be confusing respect to the use of initial data and should be better explained.

THe esclusion of readmission need a better clinical explanation of the choice.

In table 2 the statistics should be declared (median or mean, confidence interval or standard deviation).

I ask if the words in table 4 are truncated to take more cases or are typing errors.

if recall and sensistivity are the same it's better to call the measue the same way in table, formula and text; eventually in the text at rows 393 could be stated as Recall (i.e. sensitivity).

Corrections and adjustments:

Thank you very much for reading our article so carefully. In our previous publication [1], the analysis of patients admitted to the ICU included unstructured data from the initial clinical diagnosis made by the patient's physician at the time of admission. Due to the consideration of the attributes of medical text reports, it often happens that the NOTES of medical staff only adjusts the numbers, and the content is too repetitive. Therefore, we only used the physician's first diagnosis record at the time of patient admission. And we use basic vital signs in the selection of structured data. The results of statistical predictive power derived from basic vital signs and simple demographic data age are the most resource-efficient and useful. Vital signs were chosen as features mainly because most vital signs can be easily measured using non-invasive equipment, and vital signs are the most basic indicators of health that are easily understood by all healthcare professionals. The results show that our research has achieved good results.

Because the same patient might have been admitted to the ICU several times, for patients with multiple ICU admissions during a single hospitalization, only the first ICU admission record was used for analysis.

  1. Avoid double counting: If a patient has multiple hospitalizations, if each hospitalization is included in the analysis, it may lead to double counting of some data, thereby affecting the accuracy of the results.
  2. Determining causation: The cause of the first hospitalization is usually serious and well-defined, and can have a significant impact on the patient's subsequent medical decisions. Therefore, taking the first hospitalization as an object of study makes it easier to determine causality without interference from the reasons for subsequent hospitalizations.
  3. Simplified analysis: Only the first hospitalization data is used, which can simplify the analysis process and reduce the amount of data, making it easier to perform statistical analysis and model building.

In addition, in line 137-150 we mainly introduce the MIMIC-III database. Table 1: shows the MIMIC-III database compilation. And in line 180-187 are the data we used. Table 2: displays the statistical values of the eight structural variables utilized in the study. In Table 2, we present the basic descriptive statistics of this study, as well as the mean and standard deviation of the eight variables we screened in different cohorts.

In table 4, topics and keywords for dataset, the presented keywords are derived from the original data of the database, and the best topics are grouped into ten groups, each topic has twenty keywords.

We have reworked for the performance evaluation “Recall”. Thank you so much.

  •  

REsults and methods should be discussed more in depth.

Corrections and adjustments:

Thanks for your valuable suggestion. As noted by the reviewers, we have revised the Conclusions and Discussion to make the article more complete. Thank you so much.

Thank you again for taking the time out of your busy schedule to provide valuable suggestions for our articles. These suggestions lead us to revisit the original article for inaccuracies and completeness. Because of your suggestions, the article became more complete, clear and readable. Thank you so much.

  1. Chiu, C.-C., et al. Predicting the Mortality of ICU Patients by Topic Model with Machine-Learning Techniques. in Healthcare. 2022. MDPI.

Reviewer 4 Report

It is an interesting topic. It is notorious that researchers dominate the field. The methodology is well explained. Figure 4 is an excellent presentation of the results. I only have a few observations that I think are important:

1.      There are several abbreviations that do not have the full meaning, for example, what does GSC mean? (I figured it out later, but as a rule, you need to specify the long form when using the abbreviation for the first time). Precision = PPV. What does PPV mean?  In the performance evaluation of a diagnostic test, the formula TP divided by TP+FP corresponds to Sensitivity. I believe the abbreviation PPV is wrongly used here, please review it. Recall = TPR. What does TPR mean? the formula TP divided by TP+FN corresponds to the positive predictive value.

2.      Table 2,  features correspond to the first 24 hours, please specify

3.      Tables 9-11. It would help the reader to know how to interpret the score. For example,  the higher the score, the greater the precision and the recall, etc. This can be specified at the bottom of the tables. Or, it should be mentioned in the material and methods section.

Author Response

Response to Reviewer #4

We would like to thank you for your valuable recommendations that make the paper more complete and readable. We have made the following adjustments based on your suggestions.

It is an interesting topic. It is notorious that researchers dominate the field. The methodology is well explained. Figure 4 is an excellent presentation of the results. I only have a few observations that I think are important:

  1. There are several abbreviations that do not have the full meaning, for example, what does GSC mean? (I figured it out later, but as a rule, you need to specify the long form when using the abbreviation for the first time). Precision = PPV. What does PPV mean? In the performance evaluation of a diagnostic test, the formula TP divided by TP+FP corresponds to Sensitivity. I believe the abbreviation PPV is wrongly used here, please review it. Recall = TPR. What does TPR mean? The formula TP divided by TP+FN corresponds to the positive predictive value.

Corrections and adjustments:

Thank you so much for your valuable suggestions. As the reviewer pointed out, we have reworked for the some original abbreviation and the performance evaluation skills. Thank you so much.

GCS: Glasgow Coma Scale

Table Confusion matrix.

Prediction

Positive

Negative

Actual

Positive

True Positive, (TP)

False Negative, FN

Negative

False Positive, (FP)

True Negative, TN

Precision: Also called positive predictive value (PPV). This is the proportion of correct predictions in positive samples, in other words the proportion of positive samples among all positive samples classified.

Recall: Also called true positive rate (TPR). This is the proportion of samples that are predicted to be correct among factual samples, or the proportion of positive samples predicted among all positive samples.

  1. Table 2, features correspond to the first 24 hours, please specify

Corrections and adjustments:

Thanks for your valuable suggestions. That’s a very important point in this table. Thank you so much.

  1. Tables 9-11. It would help the reader to know how to interpret the score. For example, the higher the score, the greater the precision and the recall, etc. This can be specified at the bottom of the tables. Or, it should be mentioned in the material and methods section.

Corrections and adjustments:

Thanks for your valuable suggestions. We have rewritten and have modified the phrasing in "2.7 Performance Evaluation" to enhance the definition of the key points. That makes our article more clear and readable. Thank you so much.

Thank you again for taking the time out of your busy schedule to provide valuable suggestions for our articles. These suggestions lead us to revisit the original article for inaccuracies and completeness. Because of your great suggestions, the article became more complete, clear and readable. Thank you so much.
